# Polarization Engineered p-Type Electron Blocking Layer Free AlGaN Based UV-LED Using Quantum Barriers with Heart-Shaped Graded Al Composition for Enhanced Luminescence

**DOI:** 10.3390/mi14101926

**Published:** 2023-10-13

**Authors:** Samadrita Das, Trupti Ranjan Lenka, Fazal Ahmed Talukdar, Hieu Pham Trung Nguyen, Giovanni Crupi

**Affiliations:** 1Department of Electronics and Communication Engineering, National Institute of Technology Silchar, Silchar 788010, Assam, India; samadrita_rs@ece.nits.ac.in (S.D.); fazal@ece.nits.ac.in (F.A.T.); 2Department of Electrical and Computer Engineering, Texas Tech University, 1012 Boston Avenue, Lubbock, TX 79409, USA; hieu.p.nguyen@ttu.edu; 3BIOMORF Department, University of Messina, 98125 Messina, Italy; giovanni.crupi78@unime.it

**Keywords:** GaN, AlGaN, AlInN, ultra-violet (UV), electron blocking layer (EBL), light-emitting diode (LED), multi-quantum well (MQW)

## Abstract

In this paper, in order to address the problem of electron leakage in AlGaN ultra-violet light-emitting diodes, we have proposed an electron-blocking free layer AlGaN ultra-violet (UV) light-emitting diode (LED) using polarization-engineered heart-shaped AlGaN quantum barriers (QB) instead of conventional barriers. This novel structure has decreased the downward band bending at the interconnection between the consecutive quantum barriers and also flattened the electrostatic field. The parameters used during simulation are extracted from the referred experimental data of conventional UV LED. Using the Silvaco Atlas TCAD tool; version 8.18.1.R, we have compared and optimized the optical as well as electrical characteristics of three varying LED structures. Enhancements in electroluminescence at 275 nm (52.7%), optical output power (50.4%), and efficiency (61.3%) are recorded for an EBL-free AlGaN UV LED with heart-shaped Al composition in the barriers. These improvements are attributed to the minimized non-radiative recombination on the surfaces, due to the progressively increasing effective conduction band barrier height, which subsequently enhances the carrier confinement. Hence, the proposed EBL-free AlGaN LED is the potential solution to enhance optical power and produce highly efficient UV emitters.

## 1. Introduction

Due to the direct tunable band gap, between 3.43 eV and 6.11 eV, AlGaN ternary alloy is quite applicable to the fabrication of optical devices within a wavelength range of ~200–360 nm [1]. This is suitable for sterilization [2,3,4,5], ultra-violet printing [6,7], bio-medical appliances [8,9], deodorization using a photo catalyst [10,11], dermatology in medicine [12,13], and sensing applications for materials, for example urea [14,15]. However, the internal quantum efficiency (IQE) and light output power (LOP) of UV-LEDs using AlGaN alloys are still low because of several adversities. Due to the overflow of electrons at high currents, the problem of efficiency drop occurs [16]. This happens because strong induced polarization fields and quantum-confined Stark effect (QCSE) [17] lead to a significant separation of electron and hole wave functions. Thus, there is a reduction of carrier confinement and radiative recombination in the quantum well of the device.

To a certain extent, the overflow of electrons is eliminated by bringing a p-doped Al-rich electron blocking layer (EBL) between the multi-quantum well (MQW) and p-doped area [18]. However, the development of positively charged polarization sheet charges within the hetero-interface of the ultimate quantum barrier (QB) and EBL has a significant impact on the hole effectiveness [19,20]. Additionally, due to Al abundant EBL, magnesium doping efficiency is also impacted by high acceptor activation energy [21,22]. To mitigate the aforesaid challenges, various LED structures with re-designed EBL and QW have been addressed [23,24,25,26,27]. However, only a few challenges were able to be reduced, hence it is favorable to create EBL-free UV-LEDs resulting in an enhanced flow of carriers.

To tackle the difficulties caused by EBL, in this paper an EBL-free UV-LED with heart-shaped QBs has been presented operating at ~275 nm wavelength. The conventional AlGaN UV-LED consists of a thin intrinsic AlGaN strip placed into the center of each QB. In our context, this design is modified with the utilization of heart-shaped QBs on the first five barriers. The heart-shaped graded Al composition-based UV LED shows the highest quantum efficiency and reduced efficiency drop due to the reduced polarization field and the elimination of the EBL layer in the device structure resulting from the progressively graded QBs. This unique LED structure offers a significantly reduced electrostatic field in the quantum well (QW) region due to the decreased lattice mismatch between the QW and the QB. Furthermore, the internal quantum efficiency of the proposed LED exhibits a massive enhancement due to the increased carrier confinement in the device active region and the reduced electron leakage to the p-type region resulted from the progressive increase in the effective conduction band barrier heights. Moreover, the hole injection efficiency of the proposed LED structures is greatly increased due to the reduced positive polarization sheet charges at the interface of the last QB and EBL, achieved later from the EBL. Consequently, the output power and wall-plug efficiency (WPE) of the proposed LED structure show significant enhancements compared to the conventional LEDs which are about 50.4% and 49.3%, respectively.

## 2. Structural Parameters

In this study, the performance of three distinct AlGaN-based UV LED devices with non-identical MQWs and EBLs has been evaluated. To carry out the simulation, the ABC-model is used as the physical model for underlying the calculations. This model is based on the following assumptions: (i) carrier leakage from an LED structure’s active region does not result in carrier losses, (ii) active region’s non-equilibrium electron (n) and hole (p) concentrations are almost identical to one another, and (iii) carrier concentrations have a negligible impact on the recombination coefficients A, B, and C (details are explained below).

The deep ultra-violet LED fabricated by Yan et al. [28] is considered as the conventional device for reference (LED_1_) as shown in Figure 1a. The device has a dimension of 400 × 400 µm^2^ with ~275 nm wavelength emission. An AlN buffer layer and 3 µm-widen-AlGaN (60% Al; Si: 5 × 10^18^ cm^−3^) layers are settled over the substrate. The following active region comprises of 5 periods of 10 nm thick intrinsic AlGaN (50% Al) QBs and 3-nm-thick AlGaN (40% Al) quantum wells (QWs), 20 nm thick p-Al_0.65_Ga_0.35_N EBL (Mg: 2 × 10^19^ cm^−3^), succeeded by a 50 nm p-AlGaN (50% Al) cladding layer (Mg: 2 × 10^19^ cm^−3^), and finally a p-GaN contact layer of 120 nm width (Mg: 1 × 10^20^ cm^−3^). The Al composition (%) profile in the active region and EBL related to the conduction band energy diagram of LED_1_ is presented in Figure 1b.

As illustrated from the conduction band energy diagram in Figure 1c, LED_2_ is constructed from LED_1_ by replacing the uniform composition AlGaN EBL with an Al_x_In_1−x_N layer which has a graded Al composition (x: 0.65–0.35). LED_2_ includes a 3.5 nm Al_0.6_Ga_0.4_N single spike barrier and an 8.5 nm thick Al_0.5_Ga_0.5_N barrier in spite of the undoped AlGaN barriers; excluding the first barrier which is nearest to the n-doped area.

The proposed device (LED_3_) in Figure 1d is finally realized by discarding the EBL and introducing graded Al composition to each of the heart-shaped barriers, with the exception of the first one. The remaining 10 nm intrinsic QBs are now composed of graded Al (x: 0.45–0.5–0.45–0.5–0.45) to form the heart-shaped structure.

The electrical and optical characteristics of the device architectures are precisely recreated in this numerical analysis and studied with the help of commercially available industry-standard Silvaco ATLAS technology computer-aided design (TCAD).

In our work, we have used the Varshni formula to estimate the energy band gap of AlGaN as follows [29]:(1)$${\;E}_{g}\left(T\right)=E_{g}(0)-\frac{\alpha T^{2}}{\beta +T}\;$$
where *α* and *β* represent material constants, $E_{g}(0)$ and $E_{g}(T)$ denote the energy band-gap at 0, and T represents temperature, whose values are listed in Table 1 [30]. Equation (2) can be used to compute the band gap energy of $Al_{x}Ga_{1-x}N$ using these values, as shown below [31]:(2)$$E(Al_{x}Ga_{1-x}N)=x.E^{AlN}+\left(1-x\right).E^{GaN}-b.x.\left(1-x\right)$$
where *b* = 0.94 is the bowing parameter, band-offset ratio is taken as 0.68/0.35 [31]. Constructing the band gap energy or band structures of the LEDs is an important step to understand the carrier transport, electron leakage, recombination mechanism, and electrostatic fields, which are explained in the following figures. These parameters are helpful in elucidating the performance of the devices, including quantum efficiency, wall-plug efficiency, and optical and electrical properties, which are presented and explained further.

The energy band diagrams and carrier mobility are calculated using a 6×6 k.p model [32] and the Caughey–Thomas approximation [33], respectively. The Shockley-Read-Hall (SRH), Radiative, and Auger recombination co-efficients are taken as 6.67 × 10^7^/s, 2.13 × 10^−11^ cm^3^/s and 2.88 × 10^−30^ cm^6^/s, respectively [34,35], and the SRH recombination lifetime is set to 15 ns. The light extraction efficiency is considered to be 15%. Moreover, the Mg activation energy varies across 165 meV to 515 meV for p-Al_x_Ga_1−x_N alloy [22]. From the methods of Fiorentini et al., the built-in polarization (spontaneous as well as piezoelectric) is estimated [36]. The entire simulation was carried out at a temperature of 300 K. The electron and hole mobilities are set to be 100 cm^2^V^−1^s^−1^ and 5 cm^2^V^−1^s^−1^ and other band parameters are available elsewhere [37].

## 3. Results and Discussion

With the use of experimentally obtained data from standard LED_1_, the device model, parameters, and tools employed in this work were optimized. The numerical simulation of the power-current-voltage graphs of LED_1_ in Figure 2 closely matches the practically obtained curves, demonstrating the accuracy and dependability of our proposed design.

Then, numerical simulations on the electrical and optical properties of LED_1_, LED_2,_ and LED_3_ were carried out and the outcomes were compared one after another. Figure 3 illustrates the energy band diagrams (green solid lines) and quasi-Fermi levels (red dotted lines) of the three device samples at a 60 mA injected current. The variation in energy across the conduction/valence band and the quasi-Fermi levels is used to establish the efficient potential barrier heights for particles in the EBL (ϕ_E_) and QBs (ϕ_CN_). The optimum potential barrier height is acknowledged as the crucial factor in figuring out how to move the carrier. The corresponding values of ϕ_CN_ and ϕ_E_ are estimated from Figure 3a–c and listed in Table 2.

Hole injection efficiency in LED_1_ is reduced by the hole depletion zone that forms as a result of polarization charge sheet contact across EBL/LQB. The ϕ_E_ to prevent the excess of electrons in LED_1_ is 244.1 meV, which is relatively low compared to the ϕ_E_ and last QB height (ϕ_C6_) of LED_2_ and LED_3_. Due to the AlGaN spikes in the first five barriers of LED_2_, a critical polarization field is generated due to high mismatch of lattices which reduces the CBBH and could possibly escalate the electron leakage. Nevertheless, in LED_3_, the heart-shaped graded QB results are in better match of the lattice. As a result, the polarization field has dropped and remarkably raised the CBBH, thus putting forward that the final LED design can prevent electron leakage to a great extent. Also, LED_3_ shows a progressive growth in ϕ_CN_ values while gradually increasing the amount of aluminum in the QBs. As a result, the electrons are resisted from jumping out of the wells and are confined within. Therefore, the hole injection into the active region increases while non-radiative recombination in the p-region decreases. The integration of heart-shaped graded Al composition in the middle of each QB supports the idea that the negative polarization sheet charges cause the creation of regions that are being accumulated by holes. Since LED_3_ has the maximum CBBH, or ϕ_C6_, it is clear that LED_3_ is the best option for containing electrons in the active zone. But a stronger electric field is produced in the active region of LED_2_, due to an extremely large lattice imbalance within the QBs and QWs, which has an impact on carrier containment. Moreover, Table 3 calculates and provides a list of the effective VBBH ϕ_VN_ due to each QB. In comparison to LED_1_, LED_2,_ and LED_3_ have higher values of ϕ_VN_, which rises with the graded Al content of the QBs. This bolsters the enhanced hole concentration and better hole containment in the active region.

Figure 4 show the three LEDs’ carrier concentration, leak of electrons, and radiative recombination rate at 60 mA injection current. According to our analysis of Figure 4a, LED_1_ shows a crucial electronic accumulation near the LQB/EBL junction and a significantly lower electronic concentration in the MQW. Additionally, all the QWs had a greater electron concentration due to LED_3_’s noticeably higher CBBH. According to Figure 4b, LED_3_ had a higher concentration of holes in the QWs. This shows that both hole transfer and hole injecting efficiency have been improved as a result of the decreased VBBH in LED_3_.

There is a strong electron accumulation in the well and barrier regions and minimal negative electrostatic field in LED_3_, because of which electron leakage is tremendously suppressed in the p-region, as shown in Figure 4c. The gray lines represent the QWs while the pink area signifies the EBL for LED_1_ and LED_2_. With less chance of non-radiative combination of incoming holes with overflowing electrons, hole injection efficiency is increased. Also, as LED_2_ has larger electron leakage, so the non-radiative recombination increases and the hole injection efficiency decreases. 

The radiative recombination in all the QWs of LED_3_ is higher, as displayed in Figure 4d, because of better overlap of the carrier wave functions [38], minimization of electron leakage, and betterment in hole efficiency [39].

In order to have a better understanding of the physical mechanism of enhanced electron confinement in our final device LED_3_, the electrostatic field in the active region is studied mathematically. The same can be estimated using the equations marked as (3)–(5) [40].
(3)$${\;E}_{QB}\approx \frac{t_{QW}\times ∆P\left(w\right)}{t_{QW}\times \varepsilon_{QB}+t_{QB}\times \varepsilon_{QW}}\;$$
(4)$${\;E}_{QB}\times t_{QB}=E_{QW}\times t_{QW}$$
(5)$$∆P\left(w\right)=\sigma_{interface}-\rho_{QB}\times w$$$E_{QB}$ and $E_{QW}$ are the electrostatic fields in QB and QW, respectively. $∆P\left(w\right)$ denotes net polarization charge density along the growth direction ‘*w*’. $\varepsilon_{QB}$ and $\varepsilon_{QW}$ represent di-electric constants of the barrier and well, respectively, and their respective widths are $\;t_{QW}$ and ${\;t}_{QB}$. Because of the poor electrostatic field in the well, the electrons and holes are confined effectively [25]. Equation (4) depicts that $E_{QB}\;$ and $E_{QW}$ are directly proportional to each other. Also, from Equation (3), $E_{QB}$ can be minimized by decreasing the values of $∆P\left(w\right)$ which is eventually connected to $\sigma_{interface}$ and $\rho_{QB\;}$, as given in Equation (5). Again, using the following equations, the values of $\sigma_{interface}$ and $\rho_{QB}$ for LED_2_ and LED_3_ are measured and presented in Table 4. The polarization charge density at the $QB_{2}/QW_{2}$ interlayer with respect to the spontaneous ($P_{SP}$) and piezoelectric polarizations $\;(P_{PP})\;$ is defined [41]. The equations are as follows:(6)$${\;\sigma }_{interface}=\left[P_{SP}(QB)-\left\{P_{SP}(QW)+P_{PP}(QW)\right\}\right]\times 6.242\times 10^{18}$$
(7)$${\;P}_{SP}\left(Al_{x}Ga_{1-x}N\right)=-0.09x-0.034\left(1-x\right)+0.019x\left(1-x\right)$$
(8)$${\;P}_{PP}\left(Al_{x}Ga_{1-x}N\right)=\left\{x.P_{PP}\left(AlN\right)+\left(1-x\right).P_{PP}\left(AlN\right)\right\}.s\;$$
where,
$$P_{PP}\left(AlN\right)=-1.808.s+5.624.s^{2}\left(s<0\right)$$
$${\;P}_{PP}\left(AlN\right)=-1.808.s+7.888.s^{2}\left(s>0\right)$$
$${\;P}_{PP}\left(GaN\right)=-0.918.s+9.541.s^{2}$$
$$\mathrm{Basal}\;\mathrm{strain},\;s=\frac{QB_{L.C}-QW_{L.C}}{QW_{L.C}}$$

The respective lattice constants (L.C) are found elsewhere [41]. The bulk charge density in the barriers, which is induced due to polarization, can be derived as shown in Equation (9) below:(9)$${\;\rho }_{QB}=\frac{P_{SP}\left(Al_{y}Ga_{1-y}N\right)+P_{PP}\left(Al_{y}Ga_{1-y}N\right)-P_{SP}\left(Al_{x}Ga_{1-x}N\right)}{w(y)-w(x)}\times 6.242\times 10^{18}$$
where QB is graded from $Al_{x}Ga_{1-x}N$ to $Al_{y}Ga_{1-y}N$, and | w(y)−w(x)| is the grading distance. From Table 4, $\sigma_{interface\;}$ is low in LED_3_ compared to LED_2_ because of the reduced lattice mismatch at the interfaces. However, $\rho_{QB}$ values are high in LED_3_ due to the graded configuration of the Al content. As a whole, $∆P\left(w\right)$ is low in LED_3_ which leads to a poor electrostatic field in each QB. This, in turn, also reduces the electrostatic field in the QW. Figure 4c also predicts a lower electrostatic field in the MQW of LED_3_, due to which carrier confinement is enhanced in LED_3_.

The IQE, LOP, current-voltage (I-V) characteristics, and electroluminescence (EL) intensity of the three LEDs are illustrated in Figure 5. The following equations establish the structure’s current density (*j*) and it’s IQE using the ABC-model.
(10)$$\frac{j}{qd}=An+Bn^{2}+Cn^{3}$$
(11)$$IQE=\frac{Bn}{A+Bn+Cn^{2}}$$*q* = charge of electron, *A* = 1/τ (τ = Shockley–Read non-radiative carrier lifetime), *B* = radiative recombination constants, *C* = Auger recombination constants, *d* = effective width of the recombination region. The calculated maximum IQEs for LED_1_, LED_2_, and LED_3_ are 39.3%, 47.7%, and 55.3%, respectively, as shown in Figure 5a, indicating that the efficiency of LED_3_ is 40.7% and 15.9% larger than that of LED_1_ and LED_2_, respectively. Additionally, from the inset of Figure 5a, at 60 mA injection current, the efficiency drop of LED_3_ is noticeably reduced to 2.53%.

From Figure 5b, the optical power of LED_3_ is remarkably increased (~17.3 mW) at a 60 mA current, which is 61.7% better compared to the conventional device. They are attributed to improved carrier transit and confinement in the MQW, greater electron and hole wave function overlap, reduced electron leakage, and improved hole injection efficiency.

As displayed in Figure 5c, all three structures have a nearly equal turn-on voltage. It is also noticeable that at a 60 mA current, LED_3_ exhibits a slightly higher operating voltage because of the heart-shaped QBs and also due to the absence of EBL.

The constant analytical dependence of the anticipated IQE on the radiative current density (*j_rad_*) is a further significant characteristic of the ABC model, as shown in Equation (12).
(12)IQE=QQ+jradjm12+jradjm−12*Q* = quality factor which is independent of current density and equal to B(τ/C)^1/2^ within the ABC-model. Radiative recombination rates cause the value of Q to rise while the Auger and Shockley–Read recombination rates cause it to fall. Furthermore, the magnitude of Q solely determines how dependent the IQE is on the *j_rad_*/*j_m_* ratio (*j_m_* = qdB/τC within the ABC-model). As a result, this value may be used as a measure of merit for LED structures with varied designs, or those that emit light in diverse spectral regions.

According to Figure 6, the analytical dependence (12) accurately reproduces the IQE variation with current density in the structures of different LED designs. It is discovered that the specific value of the quality factor given in each plot correlates with both the maximum efficiency and its decline with current. This factor varies from 0.78 to 18, depending on how well each structure’s design and construction materials perform overall.

For clear comparison, the calculated parameters of IQE, LOP, and efficiency drop of the three samples are summarized in Table 5.

Due to better radiative recombination and lower leakage of electrons, LED_3_ displays dominant EL intensity at an emission wavelength of ~275 nm, as shown in Figure 5d. EL intensity of LED_3_ is ~1.52 times higher than LED_1_ and ~1.23 times higher than LED_2_. Peak fitting is used to calculate the full width at half maximum (FWHM) of each spectrum, as illustrated in Figure 5d. The combined impacts of the larger efficient hole injection and the better capability of heart-shaped graded Al Composition in the QBs are responsible for the superiority. Strong electron confinement in the QW of LED_3_ causes it to emit a relatively sharp and narrow light beam. As an outcome it has a small spectral width and FWHM. LED_1_, in comparison, has substantially wider radiation patterns (beam width), which result in wider FWHM and wide spectral bands.

From Figure 7a, the optical output power of LED_3_ is immensely raised along with input power supply. Also, the contrasting results of controlling the input supply in the devices is primarily ascribed to varying operational bias, as shown in Figure 5c. Figure 7b displays the wall-plug efficiency (WPE) as a factor of input current for each LED structure. The WPE of LED_3_ is ~5.39% at 60 mA, which is enhanced by nearly ~49.3% compared to LED_1_. The increased output power of our suggested device LED_3_ is what gives it a better WPE. For better understanding, the performance parameters of our work have been compared to the recent research works and their approximate values are listed in Table 6.

## 4. Conclusions

We have explored and reported the effect of UV LEDs with heart-shaped QBs and no EBLs across a wavelength range of ~275 nm. The parameters utilized are taken from the data of the experimental reference device in order to verify the validity of the simulation results. The proposed device has an emitting layer thickness of >30 nm due to which the long-life stability is 90,000 h. These simulated results reveal that incorporating the proposed QB is advantageous for achieving high optical output power and WPE in the UV spectral range. The p-EBL free device can effectively suppress the excess flow of electrons and support a boosted hole injection. The EBL-free structure is also advantageous from the perspective of epitaxial growth because it prevents the development of p-heavy doped Al composition in the EBL. This, in turn, reduces the device resistance. The proposed structure records output power of 16.1 mW and displays increased radiative recombination, IQE of 53.9% at 60 mA which is larger than the remaining structures by 1.5 and 1.8 times, respectively. Therefore, the reported LED structure shows great potential and has the ability to produce UV LEDs with significant performances for various utilities in the practical world.

## Figures and Tables

**Figure 1 micromachines-14-01926-f001:**
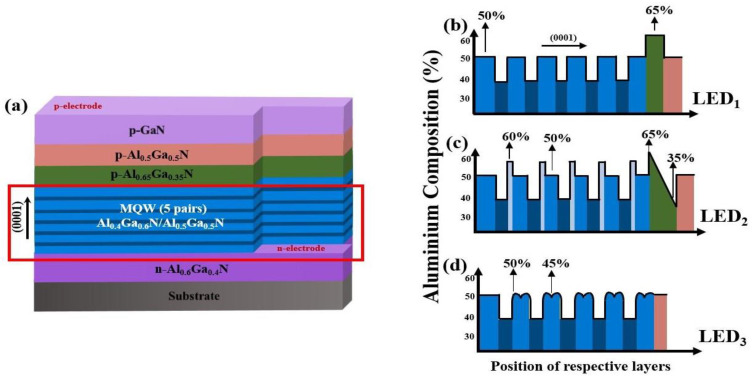
(**a**) Schematic diagrams of LED_1_, Aluminum composition (%) related to the conduction band of (**b**) LED_1_, (**c**) LED_2_, and (**d**) LED_3_.

**Figure 2 micromachines-14-01926-f002:**
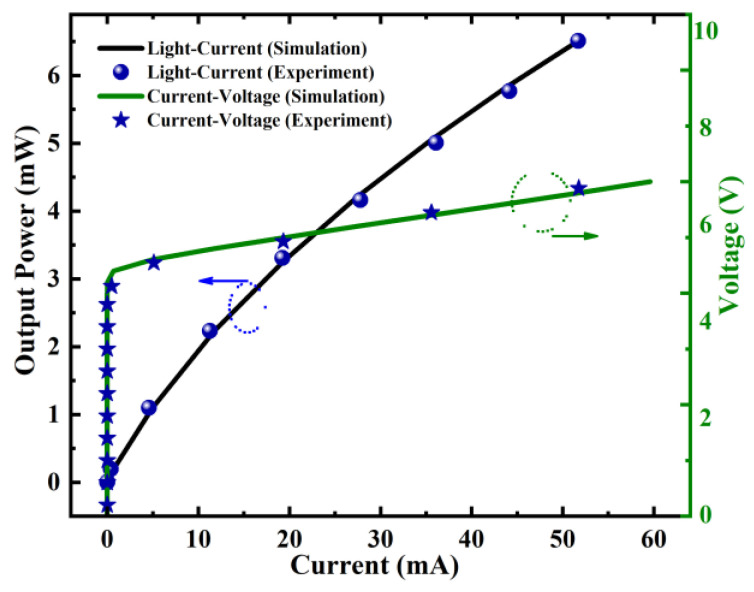
Measured and calculated light-current-voltage characteristics of LED_1_ for model validation.

**Figure 3 micromachines-14-01926-f003:**
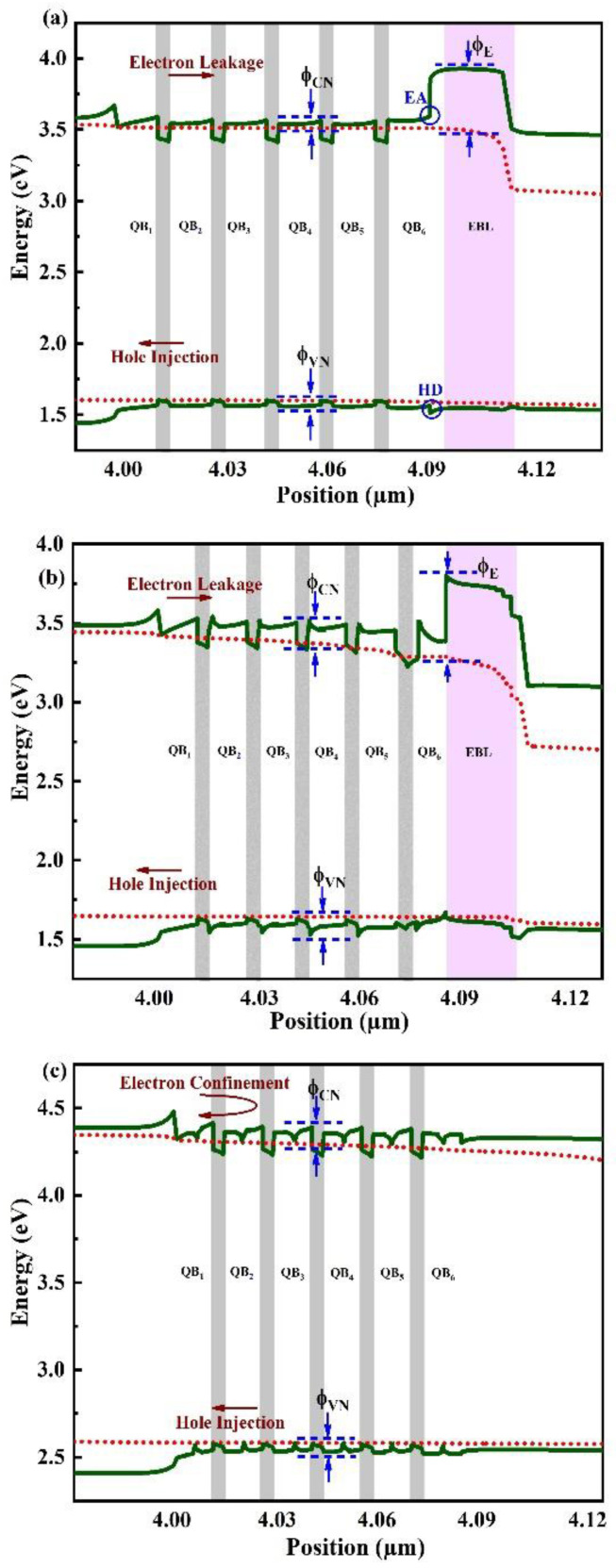
Estimated energy-band diagrams of (**a**) LED_1_, (**b**) LED_2_, and (**c**) LED_3_ at an injection current of 60 mA. E.A and H.D are the electron accumulation region in the conduction band and hole depletion region in the valence band, respectively.

**Figure 4 micromachines-14-01926-f004:**
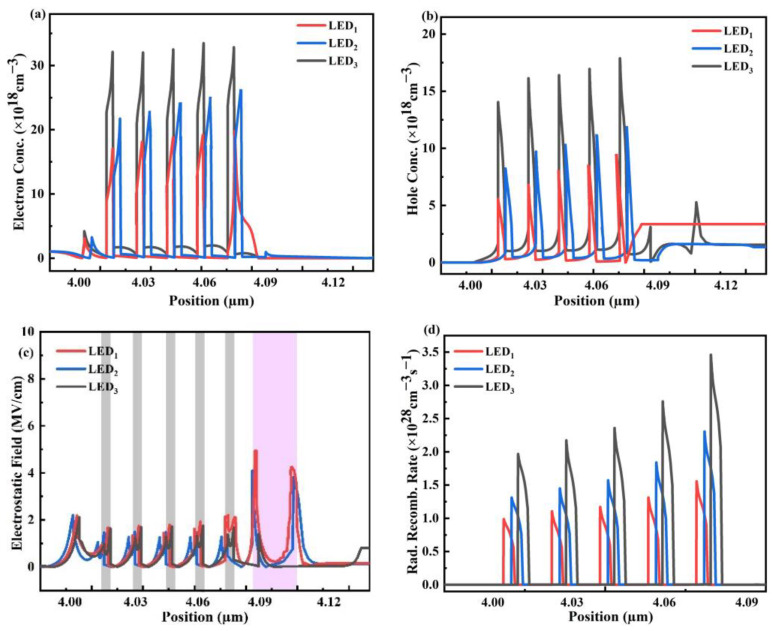
Calculated (**a**) electron concentration, (**b**) hole concentration, (**c**) electrostatic field in the active region, and (**d**) radiative rate of recombination of three structures.

**Figure 5 micromachines-14-01926-f005:**
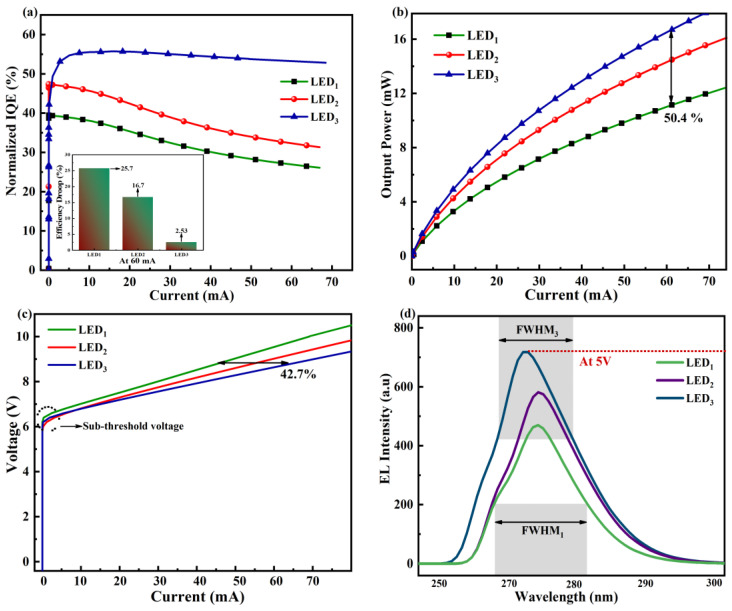
Measured (**a**) Efficiency, (**b**) Luminous Power, (**c**) I-V, and (**d**) EL intensity properties of LED_1_, LED_2_, and LED_3_.

**Figure 6 micromachines-14-01926-f006:**
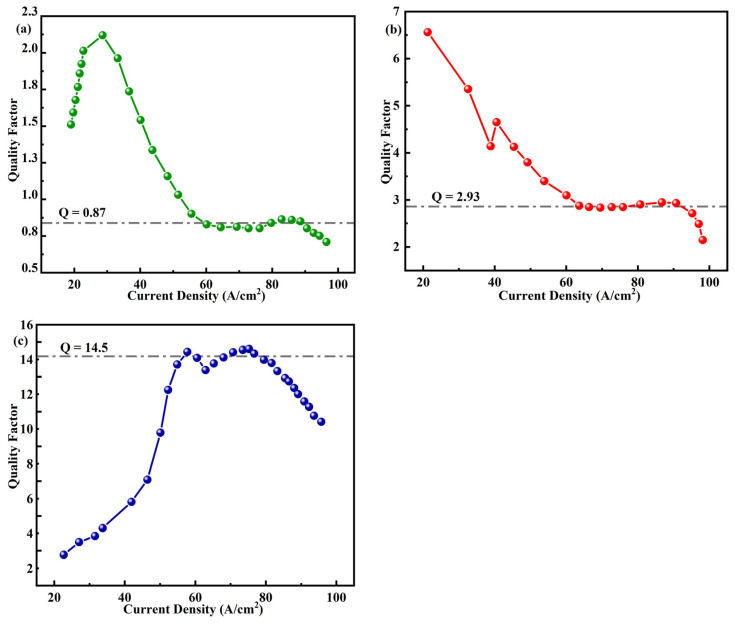
Quality factor as a function of current density obtained from the data reported for (**a**) LED_1_, (**b**) LED_2_, and (**c**) LED_3_.

**Figure 7 micromachines-14-01926-f007:**
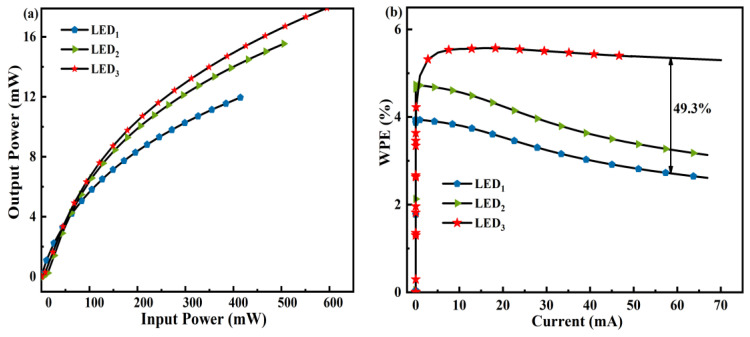
Approximated (**a**) output power with respect to input power, (**b**) WPE vs. injected current of LED_1_, LED_2_, and LED_3_.

**Table 1 micromachines-14-01926-t001:** Respective values of material constants for GaN, AlN [29].

Materials	*α*	*β*	$E_{g}(0)$
GaN	0.919 meV/K	820 K	3.507 eV
AlN	1.789 meV/K	1432 K	6.23 eV

**Table 2 micromachines-14-01926-t002:** Effective CBBH of barriers (ϕ_CN_) and EBL (ϕ_E_) for LED_1_, LED_2_, and LED_3_.

CBBH (meV)	LED_1_	LED_2_	LED_3_
ϕ_C1_	145.8	102.3	102.1
ϕ_C2_	164.4	110.7	109.1
ϕ_C3_	231.2	112.7	129.3
ϕ_C4_	298.1	115.9	194.0
ϕ_C5_	327.6	117.2	215.3
ϕ_C6_	249.8	121.4	311.9
ϕ_E_	244.1	256.0	

**Table 3 micromachines-14-01926-t003:** Effective VBBH of barriers (ϕ_VN_) for LED_1_, LED_2_, and LED_3_.

VBBH (meV)	LED_1_	LED_2_	LED_3_
Φ_V1_	225.2	302.3	252.1
Φ_V2_	224.7	317.1	261.7
Φ_V3_	223.1	329.0	272.4
Φ_V4_	222.2	362.3	289.6
Φ_V5_	221.9	372.1	294.2
Φ_V6_	220.3	384.6	309.5

**Table 4 micromachines-14-01926-t004:** Calculated $\sigma_{interface}$ at QB/QW interface, and $\rho_{QB}$ in the barriers of LED_2_ and LED_3_.

LED	$\sigma_{interface}(\times 10^{16})$				$\rho_{QB}(\times 10^{17})$			
	$QB_{2}/QW_{2}$	$QB_{3}/QW_{3}$	$QB_{4}/QW_{4}$	$QB_{5}/QW_{5}$	$QB_{2}$	$QB_{3}$	$QB_{4}$	$QB_{5}$
LED_2_	4.789	5.876	6.983	8.112	3.535	3.565	3.599	3.636
LED_3_	4.078	5.149	6.242	7.356	3.554	3.588	3.624	3.659

**Table 5 micromachines-14-01926-t005:** Optimized performance parameters for LED_1_, LED_2_, and LED_3_.

Parameters	LED_1_	LED_2_	LED_3_
Maximum IQE (%)	39.3 at 1.93 mA	47.7 at 3.28 mA	55.3 at 2.78 mA
IQE (%) at 60 mA	29.2	40.9	53.9
Efficiency Droop (%) at 60 mA	25.7	16.7	2.53
Luminous Power at 60 mA (mW)	10.7	13.9	17.3

**Table 6 micromachines-14-01926-t006:** Comparison of performance metrics with recent research works.

Author	Maximum IQE (%)	Droop (%)	Power (mW)	References
Kang et al.	27.51	3.01	8.24	[42]
Jain et al.	35.17	20.68	13.9	[43]
Velpula et al.	5.3	9.1	15.68	[44]
Ji et al.	9.16	11.24	16.11	[45]
Pandey et al.	4.12	19.21	8.91	[46]
This work	53.9	2.53	17.3	

## Data Availability

The data and materials may be available on request basis.

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
