# Peer review of "Polarization Engineered p-Type Electron Blocking Layer Free AlGaN Based UV-LED Using Quantum Barriers with Heart-Shaped Graded Al Composition for Enhanced Luminescence"

_micromachines, 2023, doi:10.3390/mi14101926_

Round 1

Reviewer 1 Report

Please explain the difference between the model in https://doi.org/10.3390/mi12030334

and this work. Is there a reason for not mentioning the same device modelling?

English is fine

Author Response

Reviewer 1:

Comments:

  1. Please explain the difference between the model in https://doi.org/10.3390/mi12030334 and this work. Is there a reason for not mentioning the same device modelling?

Our Response: The model used in our work is the ABC-model which is different from that used in the mentioned paper https://doi.org/10.3390/mi12030334. Also in LED2 (in our work), the EBL is composed of AlxIn1-xN layer having a graded Al composition (x: 0.65-0.35) which has not been studied in the referred paper https://doi.org/10.3390/mi12030334. Moreover, in the referred paper, the authors have used staircase quantum barriers (i.e. x = 0.51, 0.54, 0.57, 0.60, and 0.75, respectively for the last five barriers) whereas in our work we have used heart-shaped designed quantum barriers (i.e. x = 0.45-0.5-0.45-0.5-0.45 for each barrier except the first one). Because of the model and design used in our work, we have obtained a superior efficiency, less droop and larger luminous optical power.

Change of Place: The comparison results between the referred paper https://doi.org/10.3390/mi12030334 and our work are mentioned in Table 6 in the revised manuscript and marked in yellow color.

Comments on the Quality of English Language: English is fine.

Reviewer 2 Report

In this manuscript, Samadrita eta al reported a novel method to enhance the luminescence and decrease the electron leakage in AlGaN UV LEDs, the paper can be accepted after the following issue were concerned.

1. The stability and shape FWHM is the key process to success, what is the unique and outstandings of this recipy compare with others' work.

2. The mechanism for the high quantum yield of the LEDs should be added.

3. What is the long-life stability of this recipy?

4. The authors are suggested to add a table or figure to highlight the unique and outstanding of this compared with others

5. Progress of this field can be added. e.g.   Carbon Neutrality, 2022,1,13

English is good.

Author Response

Reviewer 2:

Comments: In this manuscript, Samadrita eta al reported a novel method to enhance the luminescence and decrease the electron leakage in AlGaN UV LEDs, the paper can be accepted after the following issue were concerned.

  1. The stability and shape FWHM is the key process to success, what is the unique and outstandings of this recipy compare with others' work.

Our Response: The answer is added in the revised manuscript.

Change of Place: Results and Discussion (Page 8), Figure (d) and marked in yellow color.

  1. The mechanism for the high quantum yield of the LEDs should be added.

Our Response: Thanks for the comment. We have included the explanation of the mechanism for high quantum efficiency of the LEDs in the Results and Discussion section, including elucidations of the band structures, carrier recombination, the enhancements of quantum efficiency, output power, and the electrical and optical properties. To make it clear and easy to follow, we added a paragraph in the revised manuscript, which can be found below:

“The heart-shaped graded Al composition-based UV LED shows a highest quantum efficiency and reduced efficiency droop due to the reduced polarization field and the elimination of the EBL layer in the device structure resulted from the progressively graded QBs. This unique LED structure offers significantly reduced electrostatic field in the quantum well (QW) region due to the decreased lattice mismatch between the QW and the QB. Furthermore, the internal quantum efficiency of the proposed LED exhibits a massively enhancement due to the increased carrier confinement in the device active region and the reduced electron leakage to the p-type region resulted from the progressive increase in the effective conduction band barrier heights.  Moreover, the hole injection efficiency of the proposed LED structures is greatly increased due to the  reduced positive polarization sheet charges at the interface of the last QB and EBL achieved from the EBL later. Consequently, the output power and wall-plug efficiency (WPE) of the proposed LED structure show significant enhancements as compared to the conventional LED which are about 50.4% and 49.3%, respectively.“

Change of Place: We added the abovementioned paragraph on page 2, lines 54-67.

  1. What is the long-life stability of this recipy?

Our Response: The answer is added in the revised manuscript.

Change of Place: Conclusion and marked in yellow color.

  1. The authors are suggested to add a table or figure to highlight the unique and outstanding of this compared with others.

Our Response: The comparison table is added in the revised manuscript.

Change of Place: Results and Discussion (Page 10, Table 6), References (Number 42-46) and marked in yellow color.

  1. Progress of this field can be added. e.g. Carbon Neutrality, 2022,1,13

Our Response: The reference is added in the updated manuscript.

Change of Place: Results and Discussion (Page 6) and References (Number 39) and marked in yellow color.

Comments on the Quality of English Language: English is fine.

Reviewer 3 Report

The authors stated that in order to solve the problem of electron leakage in AlGaN ultraviolet LEDs, they proposed an electron block-free layer of an AlGaN ultraviolet (UV) LED using polarization heart-shaped AlGaN quantum barriers (QB) instead of conventional barriers. The authors states that this novel structure has decreased the downward bend bending at the interconnection between the consecutive quantum barriers and also flattened the electrostatic field. The parameters used during simulation are extracted from the referred experimental data of conventional UV LED.

Further, the authors list some of the results they obtained as a result of modeling; no mention of the physical model underlying these calculations is even mentioned by them. The reader is invited to believe that these calculations are appropriate in the study of the above problem! In the paragraph following the introduction, the authors list the parameter sets and dimensions of three distinct AlGaN-based UV LED devices.

The authors then stated that the electrical and optical characteristics of the device architectures are accurately recreated in this numerical analysis and studied using the commercially available industry standard Silvaco ATLAS technology computer-aided design (TCAD). At the same time, not a word or half a word is said within the framework of which model these calculations were carried out and how justified is the choice of this model! At the same time, the authors provide a list of equations (1)-(2) without any explanation of the relevance of these equations! The following is a set of figures in order to convince readers that the results obtained make any sense!

Until a clear and reasonable model is proposed to describe AlGaN ultraviolet LEDs with a free layer blocking electrons using heart-shaped AlGaN quantum barriers developed by polarization instead of conventional barriers, there is no point in discussing the results further.

Author Response

Reviewer 3:

Comments: The authors stated that in order to solve the problem of electron leakage in AlGaN ultraviolet LEDs, they proposed an electron block-free layer of an AlGaN ultraviolet (UV) LED using polarization heart-shaped AlGaN quantum barriers (QB) instead of conventional barriers. The authors states that this novel structure has decreased the downward bend bending at the interconnection between the consecutive quantum barriers and also flattened the electrostatic field. The parameters used during simulation are extracted from the referred experimental data of conventional UV LED.

  1. Further, the authors list some of the results they obtained as a result of modelling; no mention of the physical model underlying these calculations is even mentioned by them. The reader is invited to believe that these calculations are appropriate in the study of the above problem!

Our Response: We apologise for the error. The physical model underlying the calculations are mentioned in the revised manuscript.

Change of Place: Introduction (Page 2), Results and Discussion (Page 7 & 8), Figure 6 and marked in yellow color.

  1. In the paragraph following the introduction, the authors list the parameter sets and dimensions of three distinct AlGaN-based UV LED devices. The authors then stated that the electrical and optical characteristics of the device architectures are accurately recreated in this numerical analysis and studied using the commercially available industry standard Silvaco ATLAS technology computer-aided design (TCAD). At the same time, not a word or half a word is said within the framework of which model these calculations were carried out and how justified is the choice of this model!

Our Response: The corrections are made in the revised manuscript.

Change of Place: Introduction (Page 2), Results and Discussion (Page 7 and 8) and marked in yellow color.

  1. Until a clear and reasonable model is proposed to describe AlGaN ultraviolet LEDs with a free layer blocking electrons using heart-shaped AlGaN quantum barriers developed by polarization instead of conventional barriers, there is no point in discussing the results further.

Our Response: The corrections are made in the revised manuscript.

Change of Place: Introduction (Page 2) and marked in yellow color.

  1. At the same time, the authors provide a list of equations (1)-(2) without any explanation of the relevance of these equations! The following is a set of figures in order to convince readers that the results obtained make any sense!

Our Response: We used these equations to calculate the bandgap energies of GaN, and AlGaN layers used in this study. Constructing the bandgap energy or band structures of the LEDs is important steps to understand the carrier transport, electron leakage, recombination mechanism and electrostatic fields, which are explained in Figures 3-4. These parameters are helpful other elucidate the performance of the devices including quantum efficiency, wall-plug efficiency, optical and electrical properties, which are presented in Figures 5-7. We have explained these equations and their relevance on page 3, from line 89 to 108.

Change of Place: N/A. 

Round 2

Reviewer 1 Report

I think having read all questions and answers that the article is ready for its readers. Best

Author Response

Response to Reviewers:

Our Response: The authors would like to thank the reviewers for their review comments and suggestions for the improvement of the manuscript. The authors are thankful to the Editor-in-Chief for giving an opportunity to showcase our state-of-the-art research in this esteemed journal.

Reviewer 1:

Comments: I think having read all questions and answers that the article is ready for its readers. Best

Our Response: The authors would like to thank the honourable reviewer.

Reviewer 2 Report

The manuscript has been improved and it can now be accepted as it is.

Author Response

Response to Reviewers:

Our Response: The authors would like to thank the reviewers for their review comments and suggestions for the improvement of the manuscript. The authors are thankful to the Editor-in-Chief for giving an opportunity to showcase our state-of-the-art research in this esteemed journal.

Reviewer 2:

Comments: The manuscript has been improved and it can now be accepted as it is.

Our Response: The authors would like to thank the honorable reviewer.

Reviewer 3 Report

In general terms, the authors have done nothing to bring the article closer to a state suitable for publication! They added a few fragments to the text that don't explain anything!

Take for example equations 1 and 2. Based on what the Varshni formula to estimate the energy band gap of  AlGaN was chosen!! Where did equation 2 come from, and what is the meaning of the combination of dimensional and dimensionless quantities on the right side of the equation 2! Below, the authors write that "The energy band diagrams and carrier mobility are calculated using 6×6 k.p  model [ 32] and Cauchy Thomas approximation [ 33] respectively. " Where does this come from and why this particular model, not some other, is the most acceptable in this case! In equations 3-9 there is a complete nonsense with dimensions! What physical mechanisms are the basis of equations 10 and 11! The same applies to equation 12!

In summary, the work does not meet the standards for articles that are published in the journal Micromachines.

Author Response

Response to Reviewers:

Our Response: The authors would like to thank the reviewers for their review comments and suggestions for the improvement of the manuscript. The authors are thankful to the Editor-in-Chief for giving an opportunity to showcase our state-of-the-art research in this esteemed journal.

Reviewer 3:

Comments:

  1. Take for example equations 1 and 2. Based on what the Varshni formula to estimate the energy band gap of AlGaN was chosen!!

Our Response: A relation for the variation of the energy gap (Eg) with temperature (T) in semiconductors is proposed:

Where α and β are constants. The equation satisfactorily represents the experimental data for diamond, Si, Ge, 6H-SiC, GaN, AlN, GaAs, InP and InAs. Where Eg is the energy gap which may be direct (Egd) or indirect (Egi); Eg(0) is its value at 0⁰K. In our present paper, we wish to suggest the above relation for the temperature dependence of the energy gap in semiconductors.

Two mechanisms which account for the majority of the fluctuation in the energy gap with temperature are mentioned below:

  • The temperature-dependent dilatation of the lattice causes a change in the relative positions of the conduction and valence bands, which demonstrates that the effect is linear with temperature at high temperatures [1, 2] . Only a small portion (about 0.25%) of the energy gap's overall change with temperature in that location can be attributed to this impact. The thermal expansion coefficient is nonlinear with temperature (T) at low temperatures, and over a range of temperatures, it even becomes negative for some solids with diamond structures [3]. In keeping with this, the dilatation's impact on the energy gap is nonlinear.
  • A shift in the relative positions of the conduction and valence bands brought on by a temperature-dependent electron lattice interaction accounts for the majority of the contribution. Theoretical treatments [4-6] show that this leads to a temperature dependence of the following form:

when

when

Where  is the Debye temperature. The values of parameters α, β and Eg(0) are calculated using the following points:

(a) Estimated from  where is the infrared resonance frequency [7].

(b) Calculated from elastic constants.

(c) Calculated from estimated values of elastic constants.

  1. Where did equation 2 come from, and what is the meaning of the combination of dimensional and dimensionless quantities on the right side of the equation 2!

Our Response: Many physical properties of  alloys can be represented as a simple analytical interpolation of the properties of its constituent compound (AC and BC) [8]. Specifically, it is found that many physical properties  of these pseudo binary alloys follow a quadratic relationship of the type:

where b represents a general bowing parameter and is approximately composition independent. Thus, the dependence of the fundamental band-gap on the mole fraction for these pseudo binary alloys is usually approximated by [9]:

where the bowing parameter b accounts for the departure from a linear behaviour. The variables A, B, and C in our work stand for Al, Ga, and N respectively. It has been demonstrated through experimentation that in pseudo binary alloys, the gap almost always bends below the straight-line average (b>0). This is explained by Schilfgaardeet et. al. [10] using a virtual crystal approximation. By breaking down the bowing parameter b into three physically separate contributions—volume deformation, various atomic electro negativities, and structural relaxation—Ferhatet et. al. describes the physical sources of the bowing [11].

For AlxGa1−xN, the experimental results for the bowing factor b range from -0.8 eV to 2.6 eV, which are summarized by Lee et al. [12]. In Ref. [12] it is shown that this difference can be related to the growth temperature of the AlGaN layers, reporting a value of 0.69 eV in the composition range 0 < x < 0.65. This value was obtained from the above equation, with the calculated band gaps of GaN and AlN.

In equation 2 (in our work), the unit of parameters on both sides is eV. The parameter x (which is dimensionless) is multiplied by each term of the right-hand side to get the influence of different aluminium compositions in the quantum barriers. Other than that all the parameters  have the same unit eV.

  1. Below, the authors write that "The energy band diagrams and carrier mobility are calculated using 6×6 k.p model [32] and Cauchy Thomas approximation [33] respectively.” Where does this come from and why this particular model, not some other, is the most acceptable in this case!

Our Response: In our work, the k.p technique is well suited to describe how strain affects the conduction band minimum. The model considers a solid as a series of equally spaced, infinitely high barriers separated by equally spaced wells which is well synchronised with our designed structure of LED.

The applications of Cauchy Thomas approximation are: (a) used to solve issues with measurement and calibration, physical anthropology, and mechanical and electrical theory, (b) It is the distribution of energy of an unstable state in quantum mechanics known as a Lorentzian distribution in physics, and (c) Also used to model the points of impact of a fixed straight line of particles emitted from a point source. These conditions suitably apply to the designing and modelling of our designed LED structures.

Hence, these models are used to calculate the energy band diagrams and carrier mobility and are the most acceptable in this case.

  1. In equations 3-9 there is a complete nonsense with dimensions!

Our Response: S.I units of the parameters used in our work are listed below:

Parameters

Symbol

Unit

Electrostatic field

 or

Newton per coulomb or volt per meter

Width

 or

Meter

Polarization charge density

Coulombs per square meter or newton per coulomb

Polarization-induced sheet charge density

Coulombs per square meter

Polarization-induced bulk charge density

Coulombs per cube meter

Using the information from above table, in equation 3 the dielectric constants have no unit. Hence, the units of equation on both sides are coulombs per square meter.

In equation 4, the units on both sides are Volt.

In equation 5, the units on both sides are coulombs per square meter.

In equation 6, the units on both sides are coulombs per square meter.

In equation 7 and 8, only the numerical values of  and  are calculated using the given equations.

In equation 9, the units on both sides are coulombs per cube meter.

  1. What physical mechanisms are the basis of equations 10 and 11! The same applies to equation 12!

Our Response: AlGaN/GaN LEDs experience a gradual drop in efficiency known as "efficiency droop" as the injection current density rises above a threshold that typically falls between 0.1 and 30 A/cm2 [13, 14]. We use the ABC model  to analyse the carrier recombination inside quantum wells (QWs) in order to thoroughly study the efficiency droop. Lack of electron capture into QWs, electron escape from QWs, the electron attracting properties of the spacer-EBL (electron blocking layer) interface, the p-type doping properties of EBL, and the asymmetry in electron and hole transport properties in GaN are some droop-causing mechanisms based on recombination outside of QWs that can result in carrier leakage out of the active region [15-18]. Carrier delocalization and Auger recombination are two recombination-based droop-inducing mechanisms that occur inside QWs (without leakage) [15, 16]. We demonstrate that the ABC + f(n) model comparison with experimental data yields insightful information regarding the efficiency droop and shows that f(n) depends on the device structure and can have second, third, and higher-order contributions.

ABC-model provides the analytical expressions for IQE. The model is criticized severely for oversimplified treatment of the considered physical processes and for some disagreement with a number of observations.

References:

[1] Moglich, R. Aud Rompe, R., Z. Phys. 119 (1942) 472.

[2] Bardeen, J. and Shockley, W., Phys. Rev. 80 (1950) 72.

[3] Collins, J. G. and White, G. K., Progress in Low Temperature Physics, 4 (North-Holland Publishing Co., Amsterdam, 1964) 450.

[4] Fan, H. Y., Phys. Rev., 78 (1950) 808.

[5] Antoncik, E., Czech. J. Phys., 5 (1955) 449.

[6] Muto, T. and Oyama, S., Prog. theor. Phys., 5 (1950) 833.

[7] Spitzcr, W. G., Kleinman, D. and Walsh, D., Phys. Rev., 113 (1959) 127.

[8] R. Nunez-Gonzalez, “First-principles calculation of the band gap of AlxGa1−xN and InxGa1−xN,” Rev. Mex. F´Isica, vol. 54, no. 2, pp. 111–118, 2008.

[9] J. E. Bernard and Alex Zunger, Phys. Rev. B36 (1987) 3199.

[10] M. Van Schilfgaarde, A. Sher, and A. B. Chen, J. Crystal Growth 178 (1997) 8.

[11] M. Ferhat and F. Bechstedt, Phys. Rev. B65 (2002) 075213.

[12] S. R. Lee et al., Appl. Phys. Lett. 74 (1999) 3344.

[13] M. H. Kim, M. F. Schubert, Q. Dai, J. K. Kim, E. F. Schubert, J. Piprek, and Y. Park, Appl. Phys. Lett. 91, 183507 (2007).

[14] T. Mukai, M. Yamada, and S. Nakamura, Jpn. J. Appl. Phys. 38, 3976 (1999).

[15] M. F. Schubert, S. Chhajed, J. K. Kim, E. F. Schubert, D. D. Koleske, M. H. Crawford, S. R. Lee, A. J. Fischer, G. Thaler, and M. A. Banas, Appl. Phys. Lett. 91, 231114 (2007).

[16] J. Piprek, Phys. Status Solidi A, Early View, 1-9 (2010).

[17] M. F. Schubert, J. Xu, Q. Dai, F. W. Mont, J. K. Kim, and E. F. Schubert, Appl. Phys. Lett. 94, 081114 (2009).

[18] M. F. Schubert, Q. Dai, J. Xu, J. K. Kim, and E. F. Schubert, Appl. Phys. Lett. 95, 191105 (2009).
